# A Study of Opiate, Opiate Metabolites and Antihistamines in Urine after Consumption of Cold Syrups by LC-MS/MS

**DOI:** 10.3390/molecules25040972

**Published:** 2020-02-21

**Authors:** Yao-Te Yen, Yin-Jue Chang, Pin-Jung Lai, Chi-Lun Chang, Ting-Yueh Chen, San-Chong Chyueh

**Affiliations:** Department of Forensic Science, Investigation Bureau, Ministry of Justice, Xindian District, New Taipei City 23149, Taiwan; b98605015@ntu.edu.tw (Y.-J.C.); r02625039@ntu.edu.tw (P.-J.L.); b98605019@ntu.edu.tw (C.-L.C.); r96524028@ntu.edu.tw (T.-Y.C.); m33096@mjib.gov.tw (S.-C.C.)

**Keywords:** codeine, morphine, heroin, chlorpheniramine, carbinoxamine, cold syrups

## Abstract

Studying the origin of opiate and/or opiate metabolites in individual urine specimens after consumption of cold syrups is vital for patients, doctors, and law enforcement. A rapid liquid chromatography–tandem mass spectrometry method using “dilute-and-shoot” analysis without the need for extraction, hydrolysis and/or derivatization has been developed and validated. The approach provides linear ranges of 2.5–1000 ng mL^−1^ for 6-acetylmorphine, codeine, chlorpheniramine, and carbinoxamine, 2.5–800 ng mL^−1^ for morphine and morphine-3-β-d-glucuronide, and 2.5–600 ng mL^−1^ for morphine-6-β-d-glucuronide and codeine-6-β-d-glucuronide, with excellent correlation coefficients (R^2^ > 0.995) and matrix effects (< 5%). Urine samples collected from the ten participants orally administered cold syrups were analyzed. The results concluded that participants consuming codeine-containing cold syrups did not routinely pass urine tests for opiates, and their morphine–codeine concentration ratios (M/C) were not always < 1. In addition, the distribution map of the clinical total concentration of the sum of morphine and codeine against the antihistamines (chlorpheniramine or carbinoxamine) were plotted for discrimination of people who used cold syrups. The 15 real cases have been studied by using M/C rule, cutoff value, and distribution map, further revealing a potential approach to determine opiate metabolite in urine originating from cold syrups.

## 1. Introduction

Opiate metabolites usually refer to the metabolites of morphine, codeine, and heroin. Morphine and codeine are natural substances present in poppies, whereas heroin is a semisynthetic product derived from morphine. Heroin, considered one of the most serious drugs abused worldwide, has led to numerous deaths—approximately 15,482 in 2017 [1]. In forensic science, several analytical methods, including gas chromatography–mass spectrometry (GC–MS) [2,3], GC–tandem MS (GC-MS/MS) [4,5], and liquid chromatography-mass spectrometry (LC-MS/MS) [6,7], have been adapted for the identification of heroin, other opiates, and their metabolites.

In humans, heroin and its byproduct 6-acetylcodeine are metabolized to morphine, 6-acetylmorphine (6-AM), morphine-3-β-d-glucuronide (M3G), morphine-6-β-d-glucuronide (M6G), codeine and codeine-6-β-d-glucuronide (C6G) [8,9]. However, some natural foods and medicines, such as cold syrups that contain certain amounts of codeine, can produce opiate metabolites in urine. Therefore, a person suspected of abusing heroin might defend himself by claiming to have taken medicines or foods containing codeine or morphine, which is referred to as the “poppy seed defense” [10,11]. As an important biomarker, 6-AM has been used for determining if a person has consumed heroin [12]. Unfortunately, owing to the short half-life of 6-AM, it can not be detected in urine samples especially after heroin consumption for 8 h [13,14]. In 1988, the US Department of Defense increased the confirmation cutoff value of morphine from 300 to 2000 ng mL^−1^ to reduce the probability of misidentification, but this led to a much lower positive confirmation rate [15,16]. In 2014, a marker for street heroin, acetylated-thebaine-4-metabolite glucuronide (ATM4G), was suggested [17]. However, ATM4G is unavailable commercially, limiting its practicality as a marker. In addition, the oral administration of poppy seeds and widely available poppy seed products were investigated [10,18,19], and they found that a morphine–codeine concentration ratio of below 1 (< 1) was a suitable index for the determination of foods containing poppy seeds [20,21]. However, few studies have examined the consumption of cold syrups, mainly because food containing opiate-ingredients caused more social problems than cold syrups in western countries [22,23,24,25]. Whereas, the opposite was true in Asia.

Codeine and antihistamines are usually mixed in cold syrups due to their pain relief and cough suppression properties in Asia [26]. Therefore, a person suspected of abusing heroin might defend himself by attributing a failed urine drug test to having taken cold syrups. To confirm this defense, ten participants were orally administered three brands of cold syrups available in pharmacy stores in Taiwan. The concentrations of opiates (morphine and codeine), opiate metabolites (M3G, M6G, C6G, and 6-AM), and the two antihistamines (chlorpheniramine or carbinoxamine) in urine samples from participants who were administered the cold syrups and the concentrations of the same compounds in samples from 15 people suspected of abusing opiates were determined through an LC-MS/MS approach. The ratios of the total concentrations of morphine and codeine, as well as distribution of the total concentration of the sum of morphine and codeine against chlorpheniramine or carbinoxamine, were all investigated to study opiate, opiate metabolites, and chlorpheniramine/carbinoxamine in urine after consumption of cold syrups.

## 2. Results and Discussion

### 2.1. Method Development and Assay Validation

Appendix A summarized the transitions, fragments, and collision energies for M3G, morphine-d_6_, morphine, M6G, C6G, codeine-d_6_, codeine, 6-AM, chlorpheniramine-d_6_, chlorpheniramine, and carbinoxamine optimized with an Agilent MassHunter optimizer. Appendix A showed an example of the successful separation of each analyte (1000 ng mL^−1^) under the optimized conditions using LC-MS/MS in extracted-ion mode, and the retention times of the analytes were in the range of 2.8–10 min.

The correlation coefficients (R^2^), calibrated linear ranges, limits of detections (LODs), carryovers, accuracies, and coefficients of variation (CV% values) for morphine, 6-AM, M3G, M6G, codeine, C6G, chlorpheniramine, and carbinoxamine were summarized in Table 1. All the calibration curves of the analytes were linear (R^2^ > 0.995) for quantitation in different linear ranges. The calibrated linear ranges were 2.5–800 ng mL^−1^ for morphine and M3G, 2.5–600 ng mL^−1^ for M6G and C6G, and 2.5–1000 ng mL^−1^ for 6-AM, codeine, chlorpheniramine, and carbinoxamine. The LODs of the analytes were 0.2–1.3 ng mL^−1^, confirming that the LC-MS/MS approach was sensitive for the quantitation of these analytes in urine samples. The linear ranges of morphine, M3G, M6G, and C6G, which contain phenol or glucuronide groups, resulting in pH-dependent ionization efficiency, were narrower than those of 6-AM, codeine, chlorpheniramine, and carbinoxamine. The LOD of morphine was higher than those of the other analytes, mainly because electrospray formation was hampered attributed to a lower content of organic modifier (water/acetonitrile = 83/17, *v*/*v*) at the retention time (3.7 min). The selectivity of this method and the influence of endogenous compounds were assessed using blank urine samples. As shown in Appendix A, unexpected interferences were not observed in the chromatograms, but obvious signals of the internal standards (ISs) and the analytes at their LODs were visible, confirming the excellent selectivity and sensitivity of the method [27,28,29,30].

Carryover was tested with the use of equation (1). The carryover indices of all analytes were within 1%, indicating the absence of significant carryover between the analyses of the working solutions at the highest and lowest tested concentrations [31]. The intraday assay precision values at low, moderate, and high concentrations of the testing analytes were 0.2%–3.6%, 0.2%–3.3%, and 0.7%–3.1%, respectively. The interday assay precision values at low, moderate, and high concentrations were 1.8%–4.2%, 1.4%–8.3%, and 0.9%–6.2% at 0, 7, and 14 days, respectively. The intraday and interday assay accuracies were in the range of 83.1% to 112.2% and 84.1% to 104.6% (i.e., within the ±20% maximum acceptance criteria) [32].

Appendix A listed the slopes and matrix effects (MEs) of the urine matrix calibration curves obtained using “dilute-and-shoot” analysis without extraction, hydrolysis and/or derivatization. The MEs were assessed by analyzing the analytes spiked into five individual blank urine samples and pure solvent according to a previously reported procedure [33,34]. Morphine, 6-AM, M6G, codeine, C6G, and carbinoxamine demonstrated MEs of 0% to 8.7%, implying that the urine matrix did not significantly affect the ionization of these analytes (within −15% to 15%) [33,35]. However, the urine matrix significantly enhanced the signals of M3G and chlorpheniramine (MEs > 15%). The highest matrix effect (approximately 32%) for M3G was observed, mainly because that higher amounts of polar constituents from urine matrix were co-eluted with M3G for ion enhancement at a short retention time (2.8 min) in a higher content of aqueous modifier (water/acetonitrile = 91/9, *v*/*v*) [36,37,38]. Thus, to reduce this effect, a matrix-match method was used for quantitation. As expected, a high precision (< 5%) of MEs was achieved in the five individual blank urine samples. The interference effects were assessed by analyzing our targets when mixed with high concentrations of the interferents. The results revealed biases in the range of −4% to 9% within ±20% of the target concentration [39]. The dilution integrity was assessed for 2-, 10- and 20-fold dilutions as shown in Appendix A. The results revealed that the accuracies were in the range of 94.0% to 106.9%, which are within the acceptable variation of ±15% of the nominal concentration, and precisions were in the range of 0.2% to 5.2% in the three replicates. Appendix A showed the excellent stabilities of the analytes during storage, the concentrations of all the analytes were stable within a variation of ±10% over a period of 28 days, indicating that urine samples stored at 4 °C prior to analysis were suitable for these assays.

### 2.2. Clinical Trials Study

Cold syrups were orally administered to ten participants following the suggested use on packaging (4.8 or 5 mg of codeine phosphate in each dose, three times a day for three days). Three different cold syrups were tested, and the average analyte concentrations and the calculated total concentration of morphine or codeine in the urine samples from the ten participants were displayed in Figure 1. For the samples (Consrine, ssBuron, and Pabron), the average concentrations of C6G (419–1190, 624–1476, and 358–1406 ng mL^−1^) and M3G (177–353, 326–508, and 114–263 ng mL^−1^) were obviously higher than those of codeine (249–940, 307–944, and 226–728 ng mL^−1^) and morphine (7.6–130, 8.7–17, and 8.2–13 ng mL^−1^), respectively, during the trial period of 0–66 h, indicating that opiate-glucuronides were the major metabolites of the opiate. When syrup consumption was terminated, the concentrations of the opiate and the opiate metabolites decreased quickly. On the other hand, chlorpheniramine or carbinoxamine was still detectable after 84–132 h of administration, and thus these compounds can be used as markers to determine the use of cold syrups. The changes in opiate concentrations as a function of time were slightly different for the three formulations due to differential lifestyles and metabolic rates of the participants, the times at which the urine samples were collected, and the small number of samples. A time interval of cold syrups taken by participants for their last dose that day until the next day’s use was approximately 12 h, which caused decreasing concentrations of each metabolite in the urine samples under 30-48 h of administration.

The details of the total concentrations of codeine or morphine in the urine samples of all participants were displayed in Figure 2. During oral administration of the cold syrups (Consrine, ssBuron, and Pabron), 14.4, 24.4, and 5.6% of the urine samples (13 of 90, 22 of 90, and 5 of 90) revealed concentrations of morphine (calculated based on the free form of morphine, M3G, and M6G) higher than 300 ng mL^−1^, but all the concentrations were under 2000 ng mL^−1^. For codeine (calculated based on the free form of codeine and C6G), 48.9, 60.0, and 46.7% of the urine samples (44 of 90, 54 of 90, and 42 of 90) revealed concentrations higher than 300 ng mL^−1^. Furthermore, 4.4, 11.1, and 8.9% of the urine samples (4 of 90, 10 of 90, and 8 of 90) showed codeine concentrations higher than 2000 ng mL^−1^. The ratios of the total concentrations of morphine and codeine (M/C) in the urine samples of all the participants were displayed in Figure 3. The results showed that the M/C values were under 1 during the trial period of 0–66 h, which was consistent with the rule of M/C < 1 [21]. Unfortunately, after 72 h of administration, the M/C values increased gradually, eventually violating the rule of M/C < 1, attributed to that the metabolite rate of morphine was slower than that of codeine [40].

Table 2 listed the concentrations of the 6-AM, opiate, opiate metabolites, and M/C values in samples from 15 real cases. A worldwide-accepted biomarker, 6-AM, is used to determine suspects for their heroin consumption. The case 1, 6, 13, and 15 found 6-AM in their urine samples, confirming those suspects for heroin use. However, 6-AM was not detected in other cases, mainly because those suspects probably did not consume heroin, or the short life of 6-AM was metabolized to morphine in bodies after heroin consumption. In addition, the M/C values and the concentrations of morphine or codeine from samples 1–8 were over 5 and over the cutoff value of 300 ng mL^−1^ (for Taiwan), respectively. Thus, we can deduce that these people did abuse heroin [20]. Interestingly, because chlorpheniramine was detected in cases 9–15, we can infer that these suspects consumed cold syrups or other chlorpheniramine-containing medicines. The M/C values from samples 9 and 10 were both within 1. As for sample 11, codeine or its metabolites were not detected, thus its M/C value referred to as infinity was close to those of clinical trial results during the trial period of 108–132 h, due to morphine being eliminated more slowly than codeine in the body [40]. The three cases (case 9–11) passing opiate urine test were not prosecuted because of their concentrations of opiate metabolites under the cutoff value (300 ng mL^−1^). On the other hand, the cases 12–15 with their M/C values over 3 as well as their concentrations of opiate metabolites over the cutoff value were all prosecuted. Based on our practical experience, cold syrups or other morphine/codeine-containing medicines were likely consumed to mask the illegal abuse of heroin, allowing the “poppy seed defense” for suspects in cases 12–15.

Furthermore, the distribution of total concentration of the sum of morphine and codeine against chlorpheniramine or carbinoxamine obtained from clinical trials, as well as those from samples 9–15 were mapped in Figure 4A,B. The results showed that distributions of samples 9–11 were overlapped with those determined in the clinical trials, confirming the suspects in the three cases consumed cold syrups containing codeine and chlorpheniramine. Conversely, samples 12-15 have no overlapping with those acquired from clinical trials, further revealing the four suspects consumed heroin and cold syrups. Finally, the suspects in cases 1–8 and 12–15 were prosecuted and thus judged in a court in Taiwan to have consumed heroin. Therefore, this distribution map showed the potential for determining opiate and opiate metabolites derived from cold syrups in urine samples. However, it was noted that for cancer patients, morphine was used for the treatment of cancer pain [41,42]. It was still difficult to discriminate against the opiate metabolites in urine from heroin or morphine without detection of 6-AM, but patients can provide certificates of diagnosis and prescriptions from doctors to prove their legal use.

## 3. Materials and Methods

### 3.1. Chemicals

Carbinoxamine, chlorpheniramine, and chlorpheniramine-d_6_ were purchased from TRC (North York, Canada). Individual solutions of 6-AM, 7-aminonitrazepam, 7-aminonimetazepam, amphetamine, benzoylecgonine, C6G, codeine, codeine-d_6_, ketamine, methamphetamine, mephedrone, 4-methylephedrine, 3,4-methylenedioxymethamphetamine hydrochloride, M3G, M6G, morphine, morphine-d_6_, and norketamine (each with a concentration of 1 mg mL^−1^) were purchased from Cerilliant (Round Rock, TX, USA). Three brands of cold syrups (Consrine, ssBuron, and Pabron) manufactured in Taiwan were purchased from local pharmacies. Consrine contains codeine phosphate (0.48 mg mL^−1^), chlorpheniramine (0.25 mg mL^−1^), *dl*-methyl ephedrine HCl (1.6 mg mL^−1^), and glycyrrhizic acid (0.25 mg mL^−1^), ssBuron contains codeine phosphate (0.5 mg mL^−1^), guaifenesin (3.33 mg mL^−1^), chlorpheniramine maleate (0.14 mg mL^−1^), and caffeine (1.33 mg mL^−1^), Pabron contains codeine phosphate (0.48 mg mL^−1^), *dl*-methyl ephedrine HCl (2.5 mg mL^−1^), and carbinoxamine maleate (0.4 mg mL^−1^). Formic acid and methanol (high-performance LC grade) (≥ 98%) were purchased from J.T. Baker (Pittsburgh, PA, USA). Ultrapure water (18.2 mΩ·cm) obtained from a Milli-Q ultrapure water system (Millipore, Billerica, MA, USA) was used in this study.

### 3.2. Urine Samples

#### 3.2.1. Clinical Trial Urine Samples

Ten participants (5 male and 5 female) aged 22–56 years with a mean weight of 67 kg were administered 10 mL of cold syrup after each meal three times a day for three days (72 h) according to suggested use described by the manufacturer. Urine samples were collected before and after oral administration of the cold syrup (i.e., before administration and 6, 18, 30, 42, 54, 66, 84, 108, and 132 h after administration), and the samples were stored at 4 °C until analysis. All participants agreed this study and gave their informed consents before experiments. All experiments were performed in compliance with the ICH E6 Guidance for Industry (E6 Good Clinical Practice: Consolidated Guidance) and approved by the Joint Institutional Review Board of Taiwan (Certificate of Approval JIRB No: 18-S-004-1).

#### 3.2.2. Urine Samples from Real Cases

Urine samples from 15 suspects in real cases involving opiate abuse were acquired from the Taipei District Prosecutors Office in Taiwan and were used for analysis. All urine samples were stored at 4 °C until analysis. According to the Court’s Judgments, 12 of the suspects were convicted of heroin consumption, and the others were found innocent.

### 3.3. Preparation of Solutions

All standard solutions (1 mg mL^−1^) were diluted with methanol to prepare stock solutions: 100 ng mL^−1^ for morphine, morphine-d_6_, 6-AM, codeine, codeine-d_6_, chlorpheniramine, carbinoxamine, and chlorpheniramine-d_6_, 1000 ng mL^−1^ for M3G, M6G, and C6G. These stock solutions were then used to optimize the transitions, fragmentors, and collision energies for these analytes using an LC-MS/MS approach. Two mixed stock solutions containing morphine, 6-AM, M3G, M6G, codeine, C6G, carbinoxamine, and chlorpheniramine (each 1000 ng mL^−1^) were prepared in a 1:1 (*v*/*v*) mixture of methanol and water for either calibration or quality control (QC). The working solutions used for calibration were prepared by diluting the mixed stock solutions with a 1:1 (*v*/*v*) mixture of methanol and water to concentrations of 2.5, 5, 10, 40, 80, 200, 400, 600, 800, and 1000 ng mL^−1^. A stock internal standard mixed solution of morphine-d_6_, codeine-d_6_, and chlorpheniramine-d_6_ were prepared with each component at a concentration of 100 ng mL^−1^ in a 1:1 (*v*/*v*) mixture of methanol and water. Similarly, QC solutions from low to high concentrations were prepared by diluting the other mixed stock solution with a 1:1 (*v*/*v*) mixture of methanol and water to concentrations of 20, 200, and 600 ng mL^−1^ for morphine and M3G, 20, 80, and 400 ng mL^−1^ for M6G and C6G, and 20, 400, and 800 ng mL^−1^ for codeine, 6-AM, carbinoxamine, and chlorpheniramine. For selectivity assessment, a mixed solution of morphine, 6-AM, M3G, M6G, codeine, C6G, carbinoxamine, and chlorpheniramine in a 1:1 (*v*/*v*) mixture of methanol and water with concentrations of their LOQ was prepared. For dilution integrity assessment, 2-, 10-, and 20-fold of low, medium, and high QC concentrations of morphine, 6-AM, M3G, M6G, codeine, C6G, carbinoxamine, and chlorpheniramine in a 1:1 (*v*/*v*) mixture of methanol and water with were separately prepared. The samples for assessing interferents were prepared by mixing morphine, 6-AM, M3G, M6G, codeine, C6G, carbinoxamine, and chlorpheniramine (each 20 ng mL^−1^) with likely interferents, including methamphetamine, amphetamine, ketamine, norketamine, 3,4-methylenedioxymethamphetamine, mephedrone, 4-methylephedrine, benzoylecgonine, 7-aminonitrazepam, and 7-aminonimetazepam (each 1000 ng mL^−1^) in a 1:1 (*v*/*v*) mixture of methanol and water.

### 3.4. Apparatus

The LC-MS/MS system consisted of an autosampler (Agilent G7167A, Santa Clara, USA), a binary pump (Agilent G7112B, Santa Clara, USA), a separation column (3.0 × 100 mm, 2.7 μm, Poroshell 120SB-AQ Agilent, Santa Clara, USA), and a mass spectrometer (Agilent 6470 QqQ-MS, Santa Clara, USA), and the urine samples were assessed in electrospray ionization mode. Mass spectrometry was performed using multiple reaction monitoring (MRM) mode to analyze 8 species, including morphine, 6-AM, M3G, M6G, codeine, C6G, chlorpheniramine, and carbinoxamine. Morphine-d_6_, codeine-d_6_, and chlorpheniramine-d_6_ were used as internal standards. The LC-MS/MS system was operated under the following conditions: drying gas temperature of 280 °C, flow rate of 8 L min^−1^, and nebulizer pressure of 45 psi. Two product ions were obtained for each analyte, and their MRM ion ratios calculated from the peak areas of the two product ions were used for qualitative evaluation, the fragment with the greater area was used for quantitation. The mobile phase was a mixture of solvent A (0.1% formic acid aqueous solution) and solvent B (100% acetonitrile) at a flow rate of 0.3 mL min^−1^. The gradient program was set as follows: 0–0.5 min (a volume ratio of solvent A to B of 97:3), 0.5–3 min (decreasing to 90:10), 3–5 min (decreasing to 70:30), 5–10 min (decreasing to 50:50), 10–15 min (decreasing to 20:80), 15–17 min (decreasing to 0:100), 17–18 min (0:100), and 18–28 min (97:3). The sample was injected using an autosampler, and the injection volume was 5 μL. The Software (Version B.08.00, Agilent Technologies Inc., 2016) was used for quantitative and qualitative analyses.

### 3.5. Method and Validation

#### 3.5.1. Analytical Strategy

According to the guideline of SWGTOX Standard Practices for Method Validation, the US Food and Drug Administration (FDA), and the European Medicines Agency (EMA) Guidelines on Bioanalytical Method [32,43,44], the parameters that must be validated include the dynamic range, LOQ, selectivity, precision, accuracy, carryover effect, matrix effect, interferents, dilution integrity (dilution factor: 2, 10, and 20), and stability. A rapid “dilute-and-shoot” analytical method was validated and used to determine the concentrations of opiate, opiate metabolites, chlorpheniramine, and carbinoxamine in urine samples from participants that had been orally administered cold syrup and from 15 people suspected of abusing opiates.

#### 3.5.2. Dynamic Ranges, LOQs, and LODs

Aliquots (20 μL) of the working solution, blank urine (20 μL), and mixed IS solution (20 μL) were mixed with 940 μL of the eluent (A: B = 97: 3) in 2-mL vials to prepare the samples for calibration (n = 5). A weighted (1/x) regression model was used to prepare calibration curves based on the peak area ratios of the analytes relative to the ISs. The back-calculated concentrations of the calibrators are within ±15% of the target values at all points except the LOQ. The lowest concentration of each analyte in the linear range was defined as the LOQ, and at this concentration, the calculated value was within ±20%. In addition, to determine the LOD, the most dilute working solution was diluted 1 to 10 times with a 1:1 (*v*/*v*) mixture of methanol and water. Aliquots (20 μL) of the aforementioned diluted working solutions, 20 μL of blank urine, and 20 μL of the mixed IS solution were mixed with 940 μL of the eluent (A: B = 97:3) in 2-mL vials to prepare the samples for qualitative analysis. The LOD of each analyte was set as the concentration producing a signal with an intensity 3 times that of the noise (S/N = 3, n = 7).

#### 3.5.3. Selectivity

The selectivity of the analytic method was assessed by comparing the chromatograms of ten lots of blank urine, blank urine spiked with the analytes at their LOQs, and IS-spiked blank urine. Each 20 μL of blank urine was mixed with 980 μL of the eluent (A: B = 97: 3) in 2-mL vials to prepare samples for assessing possible endogenous interferents (n = 10). Aliquots (20 μL) of a mixture of the analytes at their LOQ and 20 μL of blank urine were mixed with 960 μL of the eluent (A: B = 97: 3) in 2-mL vials to prepare samples to check the responses of analytes (n = 10). Similarly, 20 μL of blank urine and 20 μL of the IS mixed solution were mixed with 960 μL of the eluent (A: B = 97: 3) in 2-mL vials to prepare samples to check the response of each IS (n = 10).

#### 3.5.4. Accuracy and Precision

The recovery percentage from the nominal concentration and CV % from interday and intraday assays based on analysis of QC samples (n = 6) were used to assess the accuracy and precision of this method. QC samples were prepared by mixing 20 μL of QC solutions, 20 μL of blank urine, and 20 μL of IS mixture solution with 940 μL of the eluent (A: B = 97:3) in 2-mL vials.

#### 3.5.5. Carryover

The carryover was assessed by analyzing the most dilute working solution four times immediately after analyzing the most concentrated working solution three times. The carryover index was calculated as follows [31]:(1)$$\mathrm{Carryover}\;\mathrm{index}\;(\%)\;=\;\left(\frac{L_{1}-\frac{L_{3}+L_{4}}{2}}{\frac{H_{2}+H_{3}}{2}-\frac{L_{3}+L_{4}}{2}}\right)\times 100$$
where L_1_, L_3_, and L_4_ are the peak areas of the most dilute working solutions from runs 1, 3, and 4, respectively, and H_2_ and H_3_ are the peak areas of the most concentrated working solutions from runs 2 and 3, respectively.

#### 3.5.6. Matrix Effect

MEs were investigated according to the literature [30,31]. A urine matrix-free sample was prepared by mixing 20 μL of the working solution and 20 μL of the mixed IS solution with 960 μL of eluent (A: B = 97:3) before analysis. Urine matrix samples were prepared by spiking five 20-μL blank urine samples with 20 μL of working solution and 20 μL of the mixed IS solution and then diluting the samples with 940 μL of eluent (A: B = 97:3). To assess the ME on each analyte, experiments were performed at calibration levels with five blank urine samples. The MEs were evaluated by comparing the slope of the calibration curve of the matrix standards with that of the calibration curve of the neat standards using the following equation (2) [33,34]:(2)$$\mathrm{ME}\;(\%)\;=\;\left(\frac{slope\;of\;the\;urine\;matrix\;calibration\;curve}{slope\;of\;the\;neat\;calibration\;curve}-1\right)\times 100$$

#### 3.5.7. Interferences Study

Interference samples were prepared by mixing our target analytes with high concentrations (1000 ng mL^−1^) of likely interferents. These samples were analyzed in three replicates and then quantitatively evaluated for interference effects.

#### 3.5.8. Dilution Integrity

Each analyte with 2-, 10-, and 20-fold of QC concentrations in a 1:1 (*v*/*v*) mixture of methanol and water were diluted 2-, 10-, and 20-fold using a mixed eluent (A: B = 97:3), separately. Similarly, aliquots (20 μL) of the dilution sample, 20 μL of blank urine, and 20 μL of the mixed IS solution were mixed with 940 μL of the eluent (A: B = 97: 3) in 2-mL vials to prepare the samples (n = 3).

#### 3.5.9. Stability

For stability assessment, a mixture containing each analyte at 20 ng mL^−1^ was analyzed, and it was reanalyzed after 7, 14, 21 and 28 days of storage at 4 °C. Aliquots (20 μL) of the aforementioned mixture, 20 μL of blank urine, and 20 μL of mixed IS solution were mixed with 940 μL of the eluent (A: B = 97: 3) in 2-mL vials to prepare the samples (n = 6).

#### 3.5.10. Clinical Trials Study

An established method was applied to determine the concentrations of opiate, opiate metabolites, chlorpheniramine, and carbinoxamine in the urine samples from the ten participants. The ratios of the total concentrations of morphine and codeine were calculated. The clinical urine samples used for analysis were prepared by mixing 20 μL of each participant’s urine sample and 20 μL of IS mixture solution with 960 μL of the eluent (A: B = 97:3) in 2-mL vials. Since the concentrations of analytes in urine samples determined through an LC-MS/MS approach in this study were over the highest concentrations of calibration curves, clinical urine sample was then diluted 5-, 10-, and 20-fold using a mixed eluent (A: B = 97:3) before mixing with 20 μL of IS mixture solution with 960 μL of the eluent (A: B = 97:3) in 2-mL vials.

#### 3.5.11. Real Case Study

The concentrations of opiate, opiate metabolites, chlorpheniramine, and carbinoxamine in urine samples from 15 people suspected of heroin abuse were determined. Similar to the analysis of the clinical samples, the ratios of the total concentrations of morphine and codeine were calculated. The urine samples for analysis were separately prepared following a procedure similar to that used for the clinical samples. The real urine samples used for analysis were prepared by mixing 20 μL of each suspect’s urine sample and 20 μL of IS mixture solution with 960 μL of the eluent (A: B = 97:3) in 2-mL vials. While the concentrations of analytes in urine samples determined through the established method in this study were over the highest concentrations of calibration curves, real urine samples were then diluted 5-, 10-, and 20-fold using a mixed eluent (A: B = 97:3) before mixing with 20 μL of IS mixture solution with 960 μL of the eluent (A: B = 97:3) in 2-mL vials.

## 4. Conclusions

A rapid analytic method for the quantitation of opiate, opiate metabolites, and antihistamines (chlorpheniramine and carbinoxamine) in urine through LC-MS/MS was developed and validated. The results of a clinical trial revealed that people consuming cold syrups do not consistently pass opiate urine tests and the samples do not have morphine–codeine concentration ratios < 1 routinely. The distribution map of the total concentration of the sum of morphine and codeine against chlorpheniramine or carbinoxamine in the urine samples from participants were plotted, respectively, which were used to determine opiate metabolites in urine originated from cold syrups. Samples from fifteen people suspected of heroin abuse were analyzed and validated using M/C rule, cutoff value, and distribution map. To the best of our knowledge, this is the first report to discuss the concentrations of antihistamines and to study the distribution of total concentration of the sum of morphine and codeine against an antihistamine derived from cold syrup use.

## Figures and Tables

**Figure 1 molecules-25-00972-f001:**
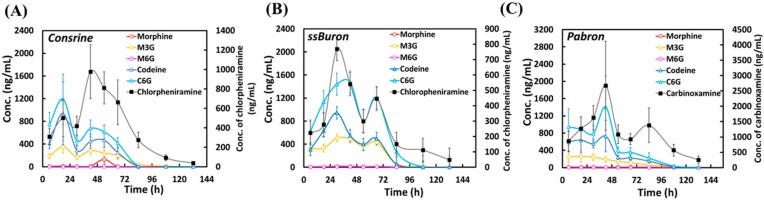
Clinical average concentrations of opiate, opiate metabolites, and antihistamines (chlorpheniramine or carbinoxamine) in the urine samples of the ten participants collecting from different times under oral administration of cold syrups (**A**) Consrine, (**B**) ssBuron, and (**C**) Pabron based on the suggested use (4.8 or 5 mg codeine phosphate for each use) from the provider.

**Figure 2 molecules-25-00972-f002:**
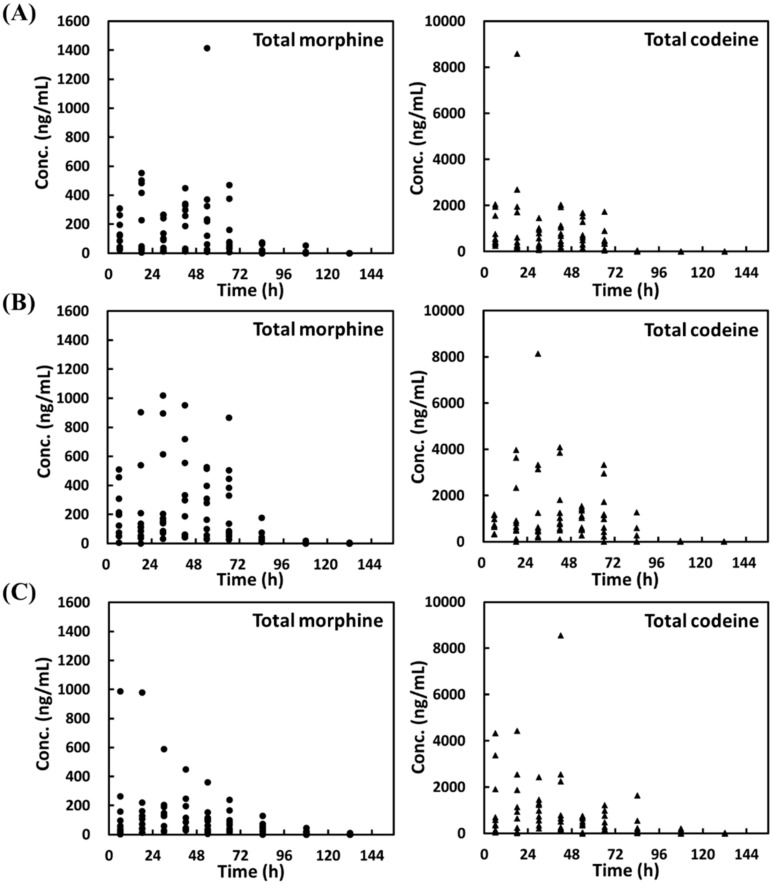
Total concentrations of morphine (left) and codeine (right) in the clinical urine samples from the ten participants collected at different times following oral administration of cold syrups based on the suggested use (4.8 or 5 mg codeine phosphate in each dose) described by the manufacturer (**A**) Consrine, (**B**) ssBuron, and (**C**) Pabron). Total morphine refers to the total concentration of morphine calculated from the free form of morphine, M3G, and M6G in the urine. Total codeine refers to the total concentration of codeine calculated from the free form of codeine and C6G in the urine.

**Figure 3 molecules-25-00972-f003:**
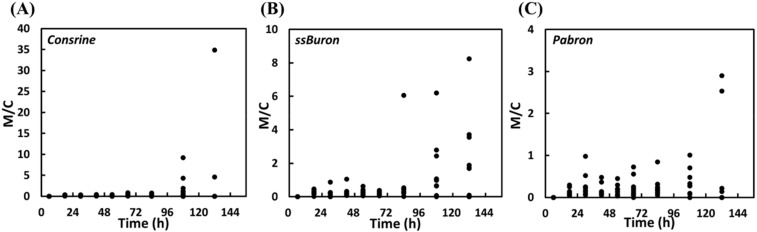
Ratios of the clinical concentrations of total morphine and total codeine (M/C) in the urine samples from the ten participants orally administered the cold syrups (**A**) Consrine, (**B**) ssBuron, and (**C**) Pabron.

**Figure 4 molecules-25-00972-f004:**
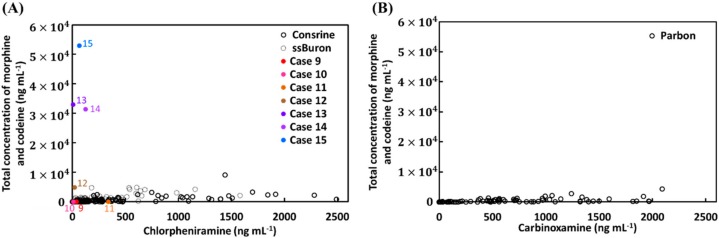
Distribution map of the total concentration of the sum of morphine and codeine against (**A**) chlorpheniramine and (**B**) carbinoxamine obtained from ten participants orally administered the cold syrups of Consrine, ssBuron, and Pabron, as well as those from seven people suspected of heroin abuse in cases 9–15.

**Table 1 molecules-25-00972-t001:** Internal standards, correlation coefficients, calibrated linear ranges, limits of detections (LODs), carryovers, accuracies, and precisions with these analytes.

Compound	Morphine	6-AM	M3G	M6G	Codeine	C6G	Chlorpheniramine	Carbinoxamine
Internal standard	Morphine-d_6_	Morphine-d_6_	Morphine-d_6_	Morphine-d_6_	Codeine-d_6_	Codeine-d_6_	Chlorpheniramine-d_6_	Chlorpheniramine-d_6_
*R* ^2^	0.998	0.998	0.997	0.999	0.997	0.998	0.999	0.999
Linear range (ng mL^−1^)	2.5–800	2.5–1000	2.5–800	2.5–600	2.5–1000	2.5–600	2.5–1000	2.5–1000
LOD (ng mL^−1^)	1.3	0.5	0.4	0.5	0.6	0.2	0.2	0.3
Carryover (%)	−0.03	0.03	0.01	0.01	−0.01	−0.03	0.01	−0.01
Accuracy for QC (%) ^a^								
Intraday (*n* = 3)								
QC-Low	89.6	88.3	89.4	86.4	89.2	83.8	83.2	83.1
QC-Medium	102.3	112.2	92.4	88.2	103.5	84.2	97.7	98.4
QC-High	108.3	110.0	99.4	93.1	102.5	88.5	100.6	101.2
Interday (*n* = 3)								
QC-Low	88.8	88.5	86.3	85.8	87.8	84.1	85.3	85.3
QC-Medium	98.9	104.6	92.7	90.2	100.9	89.0	98.0	98.4
QC-High	104.0	103.9	100.4	95.8	101.2	96.2	100.2	100.4
Precision for QC (%) ^b^								
Intraday (*n* = 3)								
QC-Low	3.6	3.0	3.2	2.8	1.1	1.7	0.6	0.2
QC-Medium	1.0	0.2	0.8	3.3	0.4	2.1	0.2	0.8
QC-High	1.8	0.9	1.7	3.1	1.3	1.3	0.7	1.5
Interday (*n* = 3)								
QC-Low	4.2	2.9	4.1	3.2	1.8	2.5	3.9	4.0
QC-Medium	4.6	6.2	3.5	8.3	2.3	7.6	1.4	1.7
QC-High	3.7	4.5	1.7	4.5	1.6	6.2	0.9	1.1

^a^ Accuracy is assessed by determining the recovery percentage of these analytes. ^b^ Precision is assessed by determining the coefficients of variation (CV % value) of these analytes.

**Table 2 molecules-25-00972-t002:** Concentrations (ng mL^−1^) and concentration ratios of opiate, opiate metabolites, and antihistamines in urine samples from 15 people suspected of heroin abuse.

Case No.	Morphine	6-AM	M3G	M6G	Codeine	C6G	Chlo.	Carb.	M/C ^b^
1	63	81	N.D. ^c^	1104	3.0	16	N.D.	N.D.	62
2	53	N.D.	14	990	81	4.1	N.D.	N.D.	8
3	73	N.D.	3116	45	23	132	N.D.	N.D.	19
4 ^a^	> 16000	N.D.	> 16000	3527	3718	2454	N.D.	N.D.	> 5
5 ^a^	11422	N.D.	> 16000	1365	1519	2024	N.D.	N.D.	> 8
6	1963	480	14339	747	792	1416	N.D.	N.D.	7
7	505	N.D.	11051	127	92	198	N.D.	N.D.	34
8 ^a^	3686	N.D.	510	> 12000	291	273	N.D.	N.D.	> 25
9	N.D.	N.D.	N.D.	N.D.	N.D.	5.6	38	N.D.	0
10	N.D.	N.D.	11	N.D.	8.9	10	8.7	N.D.	0.4
11	2.7	N.D.	N.D.	N.D.	N.D.	N.D.	337	N.D.	*∞* ^d^
12	208	N.D.	41	7182	111	89	21	N.D.	28
13 ^a^	> 16000	12	9388	> 12000	1836	3037	4.3	N.D.	> 8
14 ^a^	> 16000	N.D.	3793	> 12000	3925	2683	122	N.D.	> 5
15 ^a^	> 16000	6722	> 16000	> 12000	7029	10838	63	N.D.	> 3

^a^ Parts of analytes are over validated concentrations. ^b^ The ratios of the total concentrations of morphine and codeine (M/C). ^c^ N.D: no detection. ^d^ The ratio being referred to as infinity.

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
