# Peer review of "A Study of Opiate, Opiate Metabolites and Antihistamines in Urine after Consumption of Cold Syrups by LC-MS/MS"

_molecules, 2020, doi:10.3390/molecules25040972_

Round 1

Reviewer 1 Report

I have reviewed the manuscript, ‘A Study of Opiate, Opiate Metabolites and  Antihistamines in Urine from Cold Syrups by LC-MS/MS.’  From the yellow highlighted sections I gather that this is a revised version of a previous submission. 

The validation of the analytical method and the basic premise of the research are of a quality worthy of being included in the Journal.  However, I had some concerns about the application of the validated method to confirm heroin use.  Please see specific comments below.  Overall, the English is of a reasonable quality, though some improvement and proofreading are suggested.  I would recommend using the past tense consistently in sections such as the methods and results.

General remark:

I found the abstract confusing and lacking in structure.  A large part of the manuscript covered the development of an analytical method.  Yet, there is no mention of this in the abstract.  Consider rewriting the abstract to provide key performance indicators of the method and then give a brief description of the results relating to the m/c ratio for cold syrup users as well as the suspected drug user cohort.

Specific remarks:

Abstract, line 15.  Replace were with was

Line 18.  Replace study with studied.

Line 21.  Change to, ‘…opiate metabolites in urine…’

Introduction, line 40.  The references are rather old and do not reflect the current state of knowledge.  Modern techniques have no difficulty attaining detection levels required for 6-AM, though the short half-life is indeed a limitation, as stated.

Line 63.  The word discarnate does not appear to fit the sentence.  Do the authors mean discern or determine?

Results and Discussion, line 67-71.  It would be best if the concentrations used for the chromatograms were included in the paragraph for the readership to see the relative responses.

Line 82.  Since the morphine LOD is not orders of magnitude different from the rest, I suggest removing the sentence about its apparently poor ionization efficiency.

Line 84 and 86-90.  I have not come across papers where Authors attempt to identify interferences before developing a detection method.  It would seem to me that there are potentially a huge number of isobaric endogenous or pharmaceutical compounds (let alone pesticides and lifestyle products) and identifying a handful of illicit substances is questionable.  The chromatography and ionization patterns of the identified list of ‘suspected interferences’ should be sufficiently different that the chance of mis-identifying the target substances should be minimal.  Unless the Authors have a particular reason for insisting to keep the section, I recommend removing the passages.

Clinical trials study, line 126.  Some of the lower values for each range reported are below the method LOQ.  Please change accordingly (no values below 2.5 ng/mL can be reported).

Line 134-5.  Please remove the ‘that’ from, ‘…cold syrups that taken by participants…’, change ‘till’ to ‘until’ and change the next part of the sentence to, ‘…which caused decreasing concentrations of each…’

Line 139.  Since averages are reported in the graphs, please provide errors bars for the replicates of each point.  This will assist the readership to make sense of Fig 2.  E.g. in Fig 1 the amount of morphine and its metabolites appears to be negligible after 72 h, but from Fig 2 it is apparent that some subjects indeed had measurable amounts in their urine.

Lines 167-184.  This paragraph should start off explaining the glaringly obvious, being the absence of 6-AM in most of the samples.  This would be the best confirmation of heroin use.  Presumably the absence of 6-AM is that the samples were collected too long after the drug consumption and had been cleared from the bodies of the users.  However, the Authors need to add any insights or explanations to the discussion.  For example, if a person was on morphine medication or codeine treatment for pain (as opposed to cough syrup), wouldn’t these compounds also potentially lead to high M/C ratios and contain no 6-AM?

Lines 170-171.  The Authors state that the samples from cases 9-15 contained chlorpheniramine and therefore they were heroin users.  This does not follow intuitively.  Table 2 shows that samples 9-11 had M/C<1, which should preclude heroin use.  Please rephrase the sentence.

Line 172-175.  Sample 11 contained no codeine or its metabolite.  The Authors go on to state that this could be an example of a user that was using cough syrup to mask heroin use.  However, all results relating to cough syrups (Fig 1 and 2) contained codeine.  The results for case 11 appear to be linked neither to heroin use, nor cough syrup.  Please clarify.

Materials and method, lines 200-203 and 251-255.  As mentioned above, I do not see the need for guessing which compounds may or may not interfere.  There could be other compounds which have been missed, so why bother with such an exercise.  Do the Authors not trust their own MRM mass selections and chromatography?

Line 280.  Please state the dilution factor.

Line 323.  Please write out the full term in the heading.

Reviewer 2 Report

The manuscript entitled ”A study of opiate, opiate metabolites and antihistamines in urine from cold syrups by LC-MS/MS ” authored by Yen et al. deals with the determination of opiates in urine and how to discriminate samples from people abusing heroin from samples where the opiates instead originates from cold syrup medication.

Major comments

The title really needs to be rephrased. The urine is not from cold syrups! Furthermore, I am a bit concerned upon the ethics of the study. Have the participants, and especially those suspected for heroin abuse, given their written consent to be part of the study?

The LC-MS/MS method appears to have been properly validated. I wonder a bit though on the selectivity test for isobaric compounds. The authors claim that “the isobaric effect can be eliminated through using (sic!) multiple reaction monitoring (MRM) mode”. It is not entirely clear from the manuscript how this test has been performed, but I guess that it is more likely that the analytes of concern are separated from the added potential isobaric interferents already in the LC system. I also lack some discussion upon why M3G shows the highest ME value for a urine matrix (Table S3). Isn’t it simply because of its low retention and thereby a high risk for co-elution with polar constituents in the urine?

When it comes to the results, there are some ambiguities on the limits discussed. In the introduction, an increase in the cut-off value for morphine from 300 to 2000 ng/mL is mentioned. But in the abstract and conclusions, the authors claim that the patients (were they really patients?) consuming cold syrups containing codeine did not routinely pass the urine test for opiates, and thereby rather talk about a cutoff at 2000 ng/mL for total opiate concentration (as according to Figure 2, none of the participants in the clinical trial showed a total morphine content above the 2000-limit after cold syrup administration). And finally, when surveying the suspects, the authors are once again back at the 300 ng/mL limit as stated on line 169 and by the fact that case 2 were prosecuted and judged despite not passing the 2000 ng/mL limit. Somewhat confusing for the reader.

Finally, I think the authors should elaborate a bit more on the “distribution maps”. How useful are they really, and where should the discriminating line be drawn? Would, for example, case 12 have been on the “safe side” if he/she had taken some chlorpheniramine drug without codeine?

Minor comments

P2L46: “weight” should probably be replaced by “concentration”

P2L46: Replace “under” with “below”

P2L47: I guess one should add something to make it more clear that it is the effects of cold syrup consumption on opiate metabolites that has not been widely examined.

P2L59-63: Should really some of the results and conclusions drawn be presented here in the introduction?

P2L81: My experience is that morphine ionizes rather nicely. Couldn’t its higher LOD be an effect of the lower retention of morphine compared to codeine, and thereby a lower content of organic modifier hampering the electrospray formation when morphine elutes?

P2L90: Replace “using” with “the use of”

P6L153-154: Suggest that “M/C (<1)” is rather written as “M/C<1”

P6L154: Maybe you could add some reference and discussion upon the metabolism of codeine and its conversion to morphine, likely causing M/C values above 1 when the codeine administration is terminated?

P7L171: But chlorpheniramine is not only prescribed in combination with opiates. Couldn’t its presence be due to allergy or cough medicines for example?

Table 2: Case 10, 11/19 does not equal 0.4! Case 11, I would rather say that M/C=Infinity.

Figure 4: Why not combine these two figures into a single one with morphine+codeine vs. clorpheniramine+carbinoxamine?

P9L224: Replace “the 12” with “12 of the”

P10L262: It is not fully clear how these internal standards have been used. Only for the corresponding analyte or for clusters of analytes? This also relates to the meaning of the slopes presented in Table S3.

P12L352: “While” should probably be replaced with “Since”.

Reviewer 3 Report

This is a well-written and well-referenced article.

It is thought provoking and, although it is tested on small number of population, it could be useful in order to discern between therapeutic use of codeine and heroin abusers.

The authors drive home the message that heroin abusers could tempt to mask with use of cold syrups.

Technical issues: As second paragraphe, it should be 2. Materials and Methods with sub-paragraphes and then 3. Results and Discussion

Round 2

Reviewer 1 Report

I thank the Authors for attending to my comments and providing adequate responses.

This manuscript is a resubmission of an earlier submission. The following is a list of the peer review reports and author responses from that submission.

Round 1

Reviewer 1 Report

This manuscript provides an analytic method based on LC-MS/MS for determining whether a person consumed cold syrup or illegal abuse of heroin by using the ratio of the total concentrations of morphine and codeine over the concentration of antihistamines. Overall, the authors present a reliable method to determine opiate and opiate metabolites. There are some concerns that would require attention, which I have outlined below:

What are the concentrations of chlorpheniramine and carbinoxamine in the cold syrup? From the results of the opiate, opiate metabolites in the urine samples, the concentrations of codeine are over the range of the calibration curve, which means the calibration curve is not suitable for the quantification of the levels of opiate, opiate metabolites in the urine. The author should provide the details to prepare the real biological samples. It seems the target compounds were extracted using 97% water and 3% acetonitrile. Is it correct?

Author Response

Response letter was provided as the accessory.

Reviewer 2 Report

In the reviewed article the authors developed a method to differentiate heroin consumption and intake of cough syrups by the ratio of morphine, codeine, and metabolites over antihistamines with LC-MS/MS. The developed method was validated. The application was proofed by 15 authentic samples. The article is very well written. However, based on the following comments, the paper should be rejected.

Major Comments

Why do the authors not use accompanying alkaloids like papaverine or noscapine to differentiate between cough syrups and heroin? Base on literature, this is the usual strategy to identify a heroin abuse. The authors missed to critically discuss the classical approach, the also missed to highlight the pros and cons of the developed method.

Opiates and antihistamines are different drugs, with different metabolism and pharmacokinetics (which also include elimination steps, such as passive filtration or active secretion or reabsorption. In detail, the drugs are metabolized by different CYP enzymes, which can be affected by different inductors or inhibitors. According to literature chlorpheniramine acts as an inhibitor of Cyp2D6, which is responsible for codeine metabolism. Additionally there are well described polymorphisms for CYP enzymes, especially for Asian populations.  According to that drug and/or metabolite ratios of drugs are problematic, as each drug or metabolite may be affected by such Cyp enzyme interactions or polymorphisms. Again the authors missed to mention and critically discuss those issues with respect to recent literature.

The authors missed to discuss the situation, that a heroin abuser or drug or a patient using codeine or morphine may take an antihistamine drug. This would indeed affect the ratio of opiates over antihistamines.

Additionally no information on a single application of a cough syrup and the effect of opiate /antihistamine ratio is given. This is a crucial information, because it is more likely that if you are faking a drug test by consumption of a cough syrup, you will take a single dose (at the date of the drug test) instead of a repeated administration (“following the suggested use of packaging”).

The supplementary data is completely not available. Thus a peer review of chromatography, tuning parameters, transitions, … is not possible.

Minor

Figure 1 shows the average concentrations of opiates, opiate metabolites, and antihistamines. The unusual (elimination) curve shape should be more explained regarding repeated applications and possible creatinine normalization.

The chosen substances for selectivity are small molecules, there should be used also substances with similar chemical properties and m/z values. This could be oxycodone, oxymorphone, hydrocodone, dihydrocodeine, dextromethorphan,…

Selectivity and or chromatographic separation should also be shown for isobaric compounds such as hydromorphone (morphine) and hydrocodone (codeine). It seems that the authors completely ignored those issues.

Why do the authors not use carbnoxamine deuterated or opiates glucuronides deuterated? At least 6-MAM-glucuronide is commercially available as deuterated internal standard. The authors should at least mention the pros and cons of various IS vs. the used of only a few IS.

In my opinion, there is a typo in line 66: “Fragmentors” should be “fragments”.

At some points in the pater (e.g. line 23) “heroine” is used instead of “heroin”.

Author Response

(The authors gave the same response as above.)

Reviewer 3 Report

Please check if the reference 25 used in line 93 is a type feller. Were you trying to refer to a validation guideline or a specific published paper?

Author Response

Response:

We changed the reference 25 in our revised manuscript. (ref 25; Scientific Working Group for Forensic Toxicology. Scientific working group for forensic toxicology (SWGTOX) standard practices for method validation in forensic toxicology. J. Anal. Toxicol. 2013, 37, 452-474. 10.1093/jat/bkt054.)

Round 2

Reviewer 2 Report

After Revision, the manuscript was improoved. However, there are still some crucial issues, which are still not shown, mentioned or discussed.

Polymorphism in CYP and different elimination half lifes.

“As mentioned the different concentrations of each metabolite…are depending… o n metabolic rates…”. The answer given from the authors is one reason why you are not exspecting “statistically… a reasonable range” of concentration ratios of different drugs. As long one of the ingested drug have an alternative metabolism pathway associated to different CYP enzymes, you are not allowed to assume “statistically … a reasonable range” of concentration ratios.

Codeine is metabolized to morphine by CYP2D6. Another metabolic step is the N-demethylation to Nor-codeine by CYP3A4. Thus there are two different elimination routes with different pharmacokinetics. Image that the formation of morphine is blocked e.g. by an inhibitor or and polymorphism. Thus the codeine concentration in urine and indeed the codeine to morphine ratio is influenced.
The situation is even more complicated if you look at the concentration ratio of a second drug (e.g. c codeine / c second drug; or c morphine / c second drug), which has also its specific metabolism pathways. Each step of the elimination I. the formation of a metabolite (e.g. morphine) and II. the elimination of a drug (e.g. your antihistamine) is effected by different metabolic pathways, which can be enhanced or blocked e.g. by polymorphisms).

Thus those ratios should be carefully used and extensively discussed to cleary show the pitfalls you might be trapped in, if you use such concentration ratios.

This is especially true for any “urine” concentrations.

https://www.researchgate.net/publication/46191041_Pharmacogenetics_and_forensic_toxicology/figures?lo=1

There is still no information on this in the revised version. Expressions like “should be similar” are not a good starting point of a discussion if you are talking about forensic toxicology. There is still no information on this in the revised version. Expressions like “should be similar” are not a good starting point of a discussion if you are talking about forensic toxicology. + 7. Selectivity

The selectivity of isobaric compounds must be shown. It is not sufficient to provide literature based data on MRM transitions suggesting, which should suggest that the selected MRM transition are not effected by isobaric compounds.

If you look at the product ion spectra on nearly each of the opiates, you cleary see, that there is not one unique fragment, but a couple of (identitical) fragments present for each opiate. For that reason it is crucial to conduct those studies. As given by the authors those experiments were not conducted in the first version of the paper and those experiments were still not conducted in the revised version.

With respect to the chromatographic behavior it is not scientific to cite different retention times of compounds using a different chromatographic system. That there is a difference of retention time using different chromatographic systems you can cleary see if you look at the cited retention time of morphine (RT 4.1) and the observed one given in Table S1 (RT 3.7). The same is true for all the others compounds….

If you are diluting authentic samples due to that fact that the found concentrations are above the validated calibration range, you might do that. However, if you are doing that, you have to show in a validation experiment, that your “diluted” samples (in this case quality control sample, which are diluted) still have the right accuracy and precision. Otherwise you might also use diluted calibrators and diluted samples. Also for this second approach, validation must be conducted.